# A Systematic Review of Molecular Pathway Analysis of Drugs for Potential Use in Liver Cancer Treatment

**Ruchi Jakhmola Mani** [1,*], **Mridul Anand** [1], **Kritie Agarwal** [1], **Avi Tiwari** [1], **Qazi Amanur Rahman Hashmi** [1], **Tumul Vikram Singh** [1], **Potshangbam Nongdam** [2], **Deepshikha Pande Katare** [1,*] and **Angamba Meetei Potshangabam** [2,*]

[1]   Proteomics and Translational Research Lab, Centre for Medical Biotechnology,
      Amity Institute of Biotechnology, Amity University, Noida 201301, India; mridul.anand@s.amity.edu (M.A.);
      kritie.agarwal@s.amity.edu (K.A.); avi.tiwari@s.amity.edu (A.T.); qazi.hashmi@s.amity.edu (Q.A.R.H.);
      tumul.singh@s.amity.edu (T.V.S.)

[2]   Department of Biotechnology, Manipur University, Canchipur, Imphal 795003, India; nongpuren@gmail.com

[*]   Correspondence: rjakhmola@amity.edu (R.J.M.); dpkatare@amity.edu (D.P.K.);
      angamba@uohyd.ac.in (A.M.P.)

**Abstract:** Liver cancer is a high mortality cancer, and its increasing prevalence is a concern worldwide. Current treatment modalities for liver cancer include chemotherapy and immunotherapy. These therapies provide symptomatic relief and help prolong the lives of patients but are not an absolute cure. In this paper we have explored an alternative approach, drug repurposing, to identify drugs for treating liver cancer. Databases like PubMed, ScienceDirect, and JSTOR were used for literature mining, and the PRISMA 2020 systemic review guidelines were followed to identify drugs that have been trialed for repurposing in liver cancer. The protein receptors and target protein classes of all the drugs were identified using the Swiss Target Prediction tool. Further, the biological interactions and pathways followed by the drugs were studied via protein interaction networks using Cytoscape. Molecular pathways such as Bile acid receptor activity, Inosine-5′-monophosphate (IMP) dehydrogenase activity, JUN kinase activity, Nitric-oxide synthase activity, and Mitogen-activated protein (MAP) kinase activity were observed to be influenced by these drugs. The fact that the genes targeted by these repurposed drugs are common with the differentially expressed genes in liver cancer is an excellent starting point to verify the current hypothesis.

**Keywords:** drug repurposing; hepatocellular carcinoma; interaction networks; molecular signaling





## 1. Introduction

Worldwide, liver cancer is the second highest cause of cancer-related death and one of the few neoplasms whose incidence and mortality have been progressively growing, with the United States population experiencing the most significant risk of dying over the preceding two decades [1]. In high-risk countries like the United States, liver cancer can arise before the age of 20 years. From the year 2000, it is indicated that liver cancer remains the fifth most common malignancy in men and the eighth in women worldwide [2].

Liver cancer is a broad collection of malignant tumors that range from hepatocellular carcinoma, i.e., HCC, and intrahepatic cholangiocarcinoma, i.e., iCCA, through mixed hepatocellular-cholangiocarcinoma (HCC-CCA), fibrolamellar HCC, and the paediatric neoplasm hepatoblastoma [3,4]. Primary liver cancer is also the leading cause of cancer-related death worldwide, constituting a serious public health problem. Examples of primary liver cancer are HCC, intrahepatic iCCA, and other rare cancers such as fibrolamellar carcinoma and hepatoblastoma. The most widespread primary liver malignancies are HCC and intrahepatic cholangiocarcinoma, with other neoplasms, including combined HCC-CCA tumors, which account for fewer than 1% of cases. Liver cancer is growing worldwide, with over 1 million cases expected by 2025 [5]. With over 800,000 new cases

yearly, HCC alone accounts for 90% of all primary liver cancer cases. Because of the high frequency of hepatitis B virus (HBV) infection, Asia and Sub-Saharan Africa have the greatest incidence [6].

In contrast to other malignancies, the primary risk factors for HCC are known, including viral hepatitis (B or C), alcohol misuse, and non-alcoholic fatty liver disease in individuals with metabolic syndrome and diabetes [7–10]. iCCA, or intrahepatic cholangiocarcinoma, is the second most frequent kind of liver cancer, with the highest prevalence in Southeast Asia and the lowest incidence in Western nations. The most common kind of liver cancer is HCC, which originates in the primary type of liver cell, i.e., hepatocyte [11,12]. Like many other cancer forms, healthcare providers have more options for treating liver cancer in its early stages. Unlike many other forms of cancer, healthcare professionals understand what increases a person's risk of developing liver cancer. Healthcare professionals are working hard to identify who is more likely to acquire primary liver cancer so that it may be recognized and treated as early as possible. The distribution of liver cancer cases based on anatomical sites is shown in Figure 1.

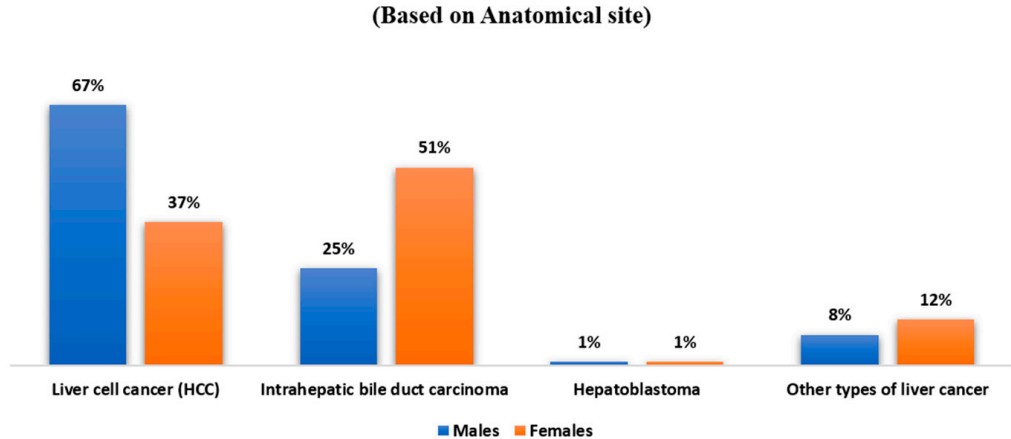

**Figure 1.** Distribution of Liver Cancer Cases based on Anatomical site.

In this paper, different drugs with the potential to treat liver cancer were explored. The drug repurposing approach is effective in introducing new drugs to the market by leveraging on the knowledge of toxicity profile, pharmacokinetics, and safety guidelines of already established drugs, thereby minimizing time and cost. Repurposed drugs have the potential to function as chemo-preventive agents and complement the effects of other chemotherapeutic drugs. They may also serve as adjuvant therapy to prevent tumor recurrence and manage the side effects of other medications. Furthermore, they can be combined with other drugs to target various oncogenic pathways or work together to eliminate the tumor completely. Nevertheless, more in-depth research is needed to fully understand their clinical properties.

### 1.1. Diagnosis, Signs, and Symptoms of Liver Cancer

There may be no apparent signs of HCC. Its symptoms include pain in the right upper abdomen, eating little, bloating, persistent tiredness, abdominal swelling, weight loss, dark urine, or yellow coloring of the eyes and skin (jaundice) [13,14]. Patients who have similar symptoms because of their underlying chronic liver disease may notice an exacerbation of these symptoms. Hepatocellular cancer is usually diagnosed by computed tomography or magnetic resonance imaging [15]. To establish the amount of liver dysfunction, blood tests are employed. A liver biopsy may be needed to confirm the diagnosis [16].

*1.2. Treatment of Liver Cancer*

1.2.1. Chemotherapy for Treating Liver Cancer

Sorafenib ($C_{21}H_{16}ClF_3N_4O_3$), an oral multikinase inhibitor, is the first-line therapy for advanced HCC [17,18]. By blocking the MAP kinase cascade and triggering apoptosis in cancer cells, this FDA-approved drug suppresses tumor angiogenesis, proliferation, and cell division. Sorafenib inhibits several proteins, including Raf-1, platelet-derived growth factor receptor, c-KIT, FLT-3, VEGF receptors -2 and -3, and RET. Regardless of the fact that patients' average survival time increased by just 3 to 5 months compared to the placebo group, the FDA approved sorafenib as a treatment for HCC in 2007 [19]. Cancer cells acquire immunity to sorafenib with repeated medication, making the therapy inefficient [20]. Moreover, when administered to cancer patients, sorafenib causes undesirable reactions. Serum lipase and amylase levels increase; so do hypertension, bleeding, neuropathy, leukopenia, lymphopenia, diarrhea, nausea, vomiting, and dyspnea. Moreover, 10% of those using sorafenib will acquire cutaneous squamous cell carcinomas [21,22]. Although sorafenib improves HCC prognosis only somewhat, recent clinical trials combine it with other drugs to produce more desirable results for patients, such as therapeutic effectiveness and fewer side effects. Vorinostat and sorafenib induce apoptosis in numerous cell lines, including HepG2 cells, by raising the activity of Bax, Bid, Bak, Bim, and Bad while lowering the activity of anti-apoptotic proteins Bcl-xl, Bcl-2, and MCL-1 [23]. Sorafenib and doxorubicin, both well tolerated by HCC, appear to benefit disease treatment. Sorafenib, which inhibits Raf-1, appears to reduce the chance of resistance development in cultured cells [24]. When compared to the two medications' individual treatments, the combination improved progression-free and overall survival. Overall, only a modest benefit is observed in HCC patients with the use of sorafenib [25]. Some of the other most common chemotherapy drugs used to treat liver cancer are gemcitabine (Gemzar) and oxaliplatin (Eloxatin). Gemcitabine is a pyrimidine analogue which is metabolized internally to its diphosphate and triphosphate forms—both of which have anti-cancer properties—by obstructing ribonucleotide reductase and competing with deoxycytidine triphosphate for DNA incorporation [26]. Cisplatin is known to act synergistically with gemcitabine and may improve disease-free and disease-specific survival in HCC patients [27]. However, cyclic therapy with gemcitabine results in an elevation of serum aminotransferase levels in 30–90% of the patients with pre-existing chronic liver disease or hepatic metastases.

1.2.2. Immunotherapy for Liver Cancer

Cancer can be treated by changing patients' immune systems so that they recognize specific antigens on cancer cells, boosting immune activity by inhibiting immunological checkpoints necessary for immunosuppressive signaling, cancer vaccines to prevent diseases or inflammatory reactions, and non-specific cancer immunotherapies that strengthen the immune system. Immunotherapy can be combined with drugs to provide a stimulatory effect, which is a benefit of this field of study [28]. For a long time, non-specific T cell activation, cytokines, and vaccination strategies have been tested in HCC, with generally unsatisfactory results [29]. However, with the FDA's approval of immune checkpoint inhibitors for the treatment of various cancers, the era of immune oncology has undergone a significant change. The Science journal named cancer immunotherapy as the innovation of the year in 2013 [30]. Immune checkpoint inhibitors target proteins that impair the capacity of the human immune system to combat the cancer cells that generate these proteins [31]. The binding of programmed cell death protein 1 and programmed cell death 1 ligand 1 to cells activates these checkpoints. PD-1, i.e., programmed cell death protein 1, is a protein that is articulated on active CD8+ and CD4+ T cells, B cells, Treg cells, natural killer cells, myeloid cells, monocytes, and progenitor cells; and PD-L1, i.e., programmed cell death 1 ligand 1, is displayed on a variety of nonimmune cells including B cells and T cells. The interaction of PD-1 and PD-L1 limits T cell activity and suppresses IFN-, interleukin-2, and other cytokines' production, resulting in temporary immune-inhibiting signals and a patient's ability to create antitumor responses that restrict cancer cell survival [32–34].

Co-inhibitory molecules are expressed by effector lymphocytes at immune checkpoints to prevent overactivation. Liver tumors and other cancers express the corresponding ligands in the tumor and stromal cells to evade anti-tumor immune responses [35]. Cytotoxic T lymphocyte-associated antigen 4 (CTLA4), which is expressed primarily by Treg cells and activated T cells, is one of the co-inhibitory receptors. It acts as an effector molecule for Treg cells and inhibits the activation of effector T cells [36]. Clinical research in the area of HCC has, thus far, concentrated on the CTLA-4 and PD-1/PD-L1 pathways. Tremelimumab, a fully human IgG2 monoclonal antibody, was the first drug to be clinically tested in HCC among CTLA-4 targeted therapies. Tremelimumab's encouraging antitumor effects in advanced HCC and its favorable safety profile in cirrhotic patients with viral causes prompted the need to test additional checkpoint inhibitors [37]. An additional mechanism of tumor-induced immune tolerance is provided by the PD-L1/PD-1 pathway. In contrast to cirrhotic patients or healthy controls, HCC patients have higher levels of PD-1 expression on effector phase CD8+T cells [38]. After hepatic resection, HCC patients who had higher levels of tumor-infiltrating and circulating PD-1+CD8+ T cells experienced earlier and more frequent disease progression. Clinical trials are being conducted in combination therapy with chemotherapeutic, immunotherapeutic, or other cancer treatment medications employing the monoclonal antibodies ramucirumab, which targets VEGF receptor-2, and bevacizumab, which inhibits VEGF receptor binding [39–42]. However, there are some risks involved in taking these medications. Patients may experience an infusion reaction. This can cause symptoms similar to an allergic reaction, such as a fever, chills, face flushing, rash, itchy skin, feeling lightheaded, wheezing, and breathing difficulties. These medications essentially disable one of the body's immune system's defenses. The lungs, intestines, liver, hormone-producing glands, kidneys, skin, and other organs can all experience serious or even life-threatening issues when the immune system begins attacking other parts of the body. Ipilimumab seems to be associated with serious side effects more frequently than PD-1 and PD-L1 inhibitors.

### 1.2.3. Common Risk Factors

Cirrhosis is a liver disease that causes scarring and increases the chance of developing HCC. Chronic hepatitis B or C infections, which are related to the greatest risk of developing HCC; extreme and persistent alcohol consumption; nonalcoholic fatty liver disease, which is predominantly related with diabetes and obesity; and other genetic liver ailments are among these conditions [43]. Because persons with hepatitis B and cirrhosis are at a higher risk of developing carcinoma, it is suggested that they have a liver ultrasound within six months.

### 1.3. Drug Repurposing

The foundation for drug repositioning is the repurposing of an active pharmacological indication [44]. Developing new therapeutic applications for previously recognized, abandoned, shelved, or experimental medicines is known as drug repurposing (drug reprofiling, indication expansion, or indication shift). Repurposing 'old' medications to treat both comparable and different diseases is becoming more appealing because it incorporates the use of derisked molecules, which may decrease total development costs and shorten research timelines [45].

Drug repositioning is based upon two scientific principles: (i) the discovery, via the elucidation of the human genome, that distinct disorders have biological targets that are sometimes shared, and (ii) the concept of pleiotropic medicines.

The approval of medical research and clinical usage takes 12 to 15 years and costs 1.2 billion dollars. Before the FDA may approve a drug for clinical use, it must have good therapeutic potential in the designated target region with low toxicity in both preclinical and clinical studies. Because of growing interest from pharmaceutical companies and observable validation of several cheminformatics and bioinformatics results, drug repurposing has

surged in prominence. Regulatory authorities have authorized around 10% of repurposed pharmaceuticals, with the other 70% in different phases of clinical testing.

Drug Repurposing Approaches

The medication repurposing approach consists of three phases before moving the potential therapy forward in the research pipeline:

- Identifying a promising chemical for a certain indication (hypothesis generation);
- Conducting a mechanistic examination of the drug's impact in preclinical models; and
- Evaluating the efficacy of a Phase II clinical research.

Computational techniques are essentially data-driven; they require a systematic examination of any type of data (such as gene expression, chemical structure, genotyping or proteomic data, or electronic health records) which may lead to the development of repurposing hypotheses. The most often used computational strategies are target-based, knowledge-based, signature-based, pathway- or network-based, and target mechanism-based [46,47]. These strategies have been shown to be both cost-effective and useful in the development of novel therapeutic drugs. Combining cheminformatics, bioinformatics, network biology, and systems biology, computational tools aid in drug development. These strategies, for example, make use of pre-existing targets, drugs, disease biomarkers, or pathways to create novel methodology and accelerate the preparation of key clinical trials.

## 2. Results and Discussion

Using the PRISMA 2020 guidelines for systematic review, a total of 16,744 entries were acquired from the PubMed, ScienceDirect and JSTOR databases after being reviewed with EndNote 20, and eliminating 223 duplicates. The remaining 16,521 records' titles and abstracts were then selected further based on their relevance to our topic of inquiry. Only 226 of these were deemed suitable, from which full-text reports were obtained. Finally, 39 papers meeting the qualifying criteria were chosen for our study. The flow diagram of this approach is given in Figure 2.

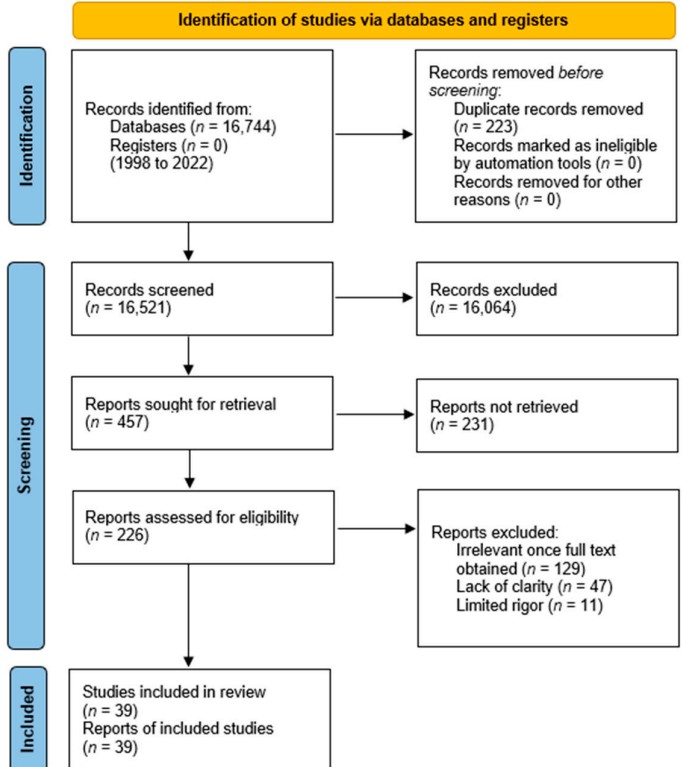

**Figure 2.** PRISMA 2020 flow diagram for systematic review.

From these studies, 14 drugs were identified from different disease pathologies which have the potential to be repurposed for liver cancer (Table 1). All of these drugs were originally developed to treat different diseases. Further analysis was carried out to determine their suitability for drug repurposing for liver cancer. The chemical structure of each drug is given in Figure 3.

**Table 1.** Details of the drugs, originally developed for different disease pathologies, identified as potential candidates for repurposing in the treatment of liver cancer.

| Drug Name | Indication | Mechanism of Action | Reference | Method of Prediction |
|---|---|---|---|---|
| Pravastatin | Lipid Lowering Drug | Inhibitor of HMG-CoA Reductase | Riaño et al. (2020) [48] | |
| Simvastatin | Lipid Lowering Drug | Inhibitor of HMG-CoA Reductase | Menter et al. (2011) [49] | |
| Fluvastatin | Lipid Lowering Drug | Inhibitor of HMG-CoA Reductase | Sławińska-Brych et al. (2014) [50] | |
| Metformin | Treat Type 2 Diabetes | Suppresses The Production of Gluconeogenic Enzymes | Rena et al. (2017) [51] | |
| Canagliflozin | Treat Type 2 Diabetes | SGLT2 Inhibitors | Luo et al. (2021) [52] | |
| Pimozide | Antipsychotic Drug | Inhibiting Dopaminergic, Serotonergic, And Unknown Central Nervous System Receptors | Kongsamut et al. (2002) [53] | |
| Valproate | Anticonvulsive Drug | Blockade of Voltage-Gated Sodium Channels And Increased Brain Levels of (GABA) | Rithanya and Ezhilarasan (2021) [54] | Target Prediction using Swiss Target tool and Protein Interaction Network analysis using STRING |
| Bexarotene | Treat Cutaneous Manifestations of T Cell Lymphomas | Targets And Activates Retinoid X Receptors (RXRs) | Qu and Tang (2010) [55] | |
| Chloroquine | Treat Malaria | Inhibition of Autophagic Flux | Solomon and Lee (2009)[56] | |
| Linagliptin | Treat Type 2 Diabetes | Inhibitor of DPP-4 | Li et al. (2020) [57] | |
| Lidocaine | Local Anaesthetics | Variety of Pathways, Including Sodium Channel Inhibitors And The Control of G Protein-Coupled Receptors | Zhou et al. (2020) [58] | |
| Raloxifene | To Treat Postmenopausal Osteoporosis | Benzothiophene Selective Estrogen Receptor Modulator | Hong et al. (2021) [59] | |
| Itraconazole | Antifungal Medicine | Inhibits Cytochrome P-450-Dependent Enzymes | Wang et al. (2020) [60] | |
| Clofazimine | Antimycobacterial And Anti-Inflammatory Agent | Increases The Activity of Bacterial Phospholipase A2 | Xu et al. (2020) [61] | |

*2.1. Pravastatin*

Pravastatin is a statin which is a competitive inhibitor of 3-hydroxy-3-methylglutaryl-coenzyme A (HMG-CoA) reductase and is used as a lipid lowering drug. Statins also have additional effects, other than their intended use as lipid lowering agents. Pravastatin is the bioactive metabolite of mevastatin which is isolated from *Penicillium citrinum.* Among all the statins, Pravastatin has the most hydrophilic property, and is freely soluble in water and ether [48]. There is an overexpression of Ras protein in the HepG2 cell line derived

from the human HCC, which is closely regulated by cholesterol metabolism. Pravastatin decreases the rate of cholesterol biosynthesis in these cell lines.

**Figure 3.** Two-dimensional chemical structure of the drugs.

*2.2. Simvastatin*

Simvastatin is also a type of statin, like Pravastatin. It is derived from *Aspergillus terreus* synthetically. Simvastatin competitively inhibits hepatic hydroxymethyl-glutaryl coenzyme A (HMG-CoA) reductase, the enzyme which catalyzes the conversion of HMG-CoA to mevalonate, a key step in cholesterol synthesis. Hydrophilic statins such as Pravastatin require the distinct expression of a transporter peptide, OATP1B1, for transfer into hepatic cells, whereas hydrophobic statins like Simvastatin are easily distributed in the cells. Simvastatin and Pravastatin both effectively slow the growth of hepatocytes that express OATP1B1. Simvastatin is more extensively incorporated into hepatocytes than Pravastatin, which is consistent with their suppression-related actions. Conversely, Simvastatin inhibits the growth of tumor cells lacking OATP1B1 which do not incorporate or respond to Pravastatin [49]. A study by Csomó et al. reported that Simvastatin can increase the oxidizing capability of free cytochrome c which, in turn, raises oxidative stress and thereby encourages apoptosis [62].

### 2.3. Fluvastatin

Fluvastatin is one of the first fully synthetic HMG-CoA reductase inhibitors. Fluvastatin has unique anti-cancer properties in addition to lowering cholesterol, such as inducing tumor cell death in several cancer cell lines (such as glioma and breast cancer cell lines) [50]. Additionally, Fluvastatin has been suggested as a possible treatment for HCC [63]. The anti-cancer property of Fluvastatin is related to its effect via the SREBP-1 and AMPKa pathway. Fluvastatin activates SIRT6 which is involved in cholesterol homeostasis. After treatment of HepG2 cells in mice, SIRT6 is activated, which leads to the deacetylation of H3K9 and H3K56 and the inhibition of lipid metabolism [64]. Fluvastatin also inhibits cholesterol synthesis via SREBP-1 phosphorylation.

### 2.4. Metformin

Metformin blocks the mitochondrial respiratory chain at the molecular level in the liver, thereby activating AMPK, improving insulin sensitivity via its effects on fat metabolism, and lowering cAMP, which, in turn, suppresses the production of gluconeogenic enzymes. Furthermore, fructose may be inhibited by AMP in the liver as a result of metformin's AMPK-independent actions on the liver [51]. Metformin has been demonstrated to reduce ATP concentration—an allosteric inhibitor of pyruvate kinase—in isolated rat hepatocytes, resulting in a reduction in glucose production via boosting pyruvate kinase flux [65]. Patients with type 2 diabetes who take metformin had a 62% lower chance of developing liver cancer according to estimates. Diabetic patients who use metformin over the long term have a decreased chance of developing cancer and a lower overall cancer death rate [66].

### 2.5. Canagliflozin

Advanced NASH increases the risks of cirrhosis and HCC, which can be countered by reducing the serum ALT baseline. Canagliflozin is a drug from the sodium-glucose cotransporter 2 (SGLT2) inhibitors class [52]. They inhibit the SGLT2 transporter in the S1 segment of the proximal tubule in the kidney and cause glycosuria and natriuresis. Their mechanism of action involves natriuresis, restoration of tubule-glomerular feedback, and amelioration of internal hypoxia. All stages of the development of liver cancer exhibit hypoxia. Hypoxia causes hypoxia-inducible factors (HIFs) to stabilize, and HIFs function as central regulators to reduce the innate immunity [67]. There are possible anti-inflammatory and antifibrotic effects for SGLT2 inhibitors as well. Canagliflozin has been proven to significantly decrease ALT levels from the baseline. It also significantly improved the hepatic fibrosis markers such as the FIB-4 index and the FM-fibro index, suggesting the possibility of improving hepatic fibrosis [68]. Canagliflozin also works on a pathway involving SGLT2 and GLUT1 which leads to the suppression of intracellular glucose uptake in HCC cells [69,70].

### 2.6. Pimozide

Pimozide, an oral active antipsychotic drug used to treat motor tics, refractory phonic tics, persistent psychosis, and Tourette's syndrome [71], has received much attention as a possible anticancer drug. This medication affects neurons in the central nervous system by inhibiting dopaminergic, serotonergic, and unknown central nervous system receptors. Due to the HERG channel affinities of pimozide, it exhibits low (10-fold) or no selectivity for D2 or 5 HT2A receptors [53]. This lack of selectivity leads to a number of secondary changes in central dopamine metabolism and function, which have both unpleasant effects as well as therapeutic implications against resistant phonic tics and the symptoms of schizophrenia and psychosis. Inhibiting stem-like cells and carcinogenesis in HCC are additional effects of pimozide (HCC). Pimozide reverses the stem-like cell tumorigenic phenotypes caused by IL-6 treatment in HCC cells and prevents the maintenance and carcinogenesis of HCC stem-like cells (CD133-positive cells). Pimozide's anticancer effects were also demonstrated in a nude mouse HCC xenograft model [72].

## 2.7. Valproate

Antineoplastic properties of the well-known anticonvulsive drug valproic acid (VPA) were discovered in 1997. The short chain fatty acids VPA and VPA analogues potently alter the biology of various types of cancer cells by promoting differentiation, reducing proliferation, increasing apoptosis and immunogenicity, and reducing angiogenesis and metastatic potential. Several studies revealed a close relationship between HDAC and the growth of malignant tumor cells and tumor cell differentiation (histone deacetylases). In cancer cells, VPA reduces the activity of the HDAC gene [73]. Multiple exogenous reporter genes, including SV40, p21, and gelsolin, which are linked to HDAC inhibition, were expressed as a result of the use of VPA and its analogues [74]. HCC cells are induced to produce NKG2DL mRNA by sodium valproate. By increasing NKG2DL expression, the HDAC-I VPA may be able to induce NK cell lysis in hepatoma cells [54].

## 2.8. Bexarotene

Bexarotene has been approved by the FDA for treating cutaneous manifestations of T cell lymphomas in a limited manner. It is a scientifically proven orphan nuclear agonist and is also a member of the retinoid subclass that specifically targets and activates retinoid X receptors (RXRs) [75]. These receptors, after activation, function as transcription factors that are involved in the regulation of expression of genes that directly influence cellular proliferation and differentiation [76]. Retinoids are biologically active vitamin A derivatives that play crucial roles in regulating cell proliferation, differentiation, and apoptosis in both embryonic and adult cell behavior. Two different families of intracellular receptors—retinoid X receptors (RXR)-$\alpha$, -$\beta$ and -$\gamma$, and retinoid acid receptors (RARs)-$\alpha$, -$\beta$ and -$\gamma$—mediate the biological effects of retinoids. Bexarotene is a selective RXR agonist that inhibits angiogenesis and metastasis while inhibiting angiogenesis and cell cycle progression, causing apoptosis and differentiation, preventing multidrug resistance, and blocking cell cycle progression [55].

## 2.9. Chloroquine

Chloroquine-based medications, primarily used to treat malaria, involve autophagy as the target mechanism which affects the inflammatory response and cancer growth. Chloroquine's pharmaceutical inhibition of autophagic flux causes an increase in apoptosis and a reduction in cell viability in hepatoma cells. When combined with presently used chemotherapy drugs, chloroquine dramatically slows tumor growth and enhances the efficacy of the drugs. Chloroquine has been found to trigger the arrest of cell cycle in the G0/G1 phase and also cause damage to the DNA. It makes the tumor cells more sensitive to the chemotherapy drugs and, hence, can be a potential repurposed therapeutic for HCC.

## 2.10. Linagliptin

Linagliptin, a hypoglycemic medication, has been shown to reduce cell proliferation by cell cycle arrest and induce apoptosis in HCT116 cells [77]. It has also been shown to inhibit tumor formation in nude mice with HCT116 cells. Linagliptin prevents cell growth in HCT 116 by causing cell cycle arrest at the G2/M and S phase and, by reducing the expression of Ki67, a nuclear protein expressed in all proliferating cells, linagliptin inhibits the growth of tumors [57]. The main mechanism of action of linagliptin to inhibit cell proliferation and promote cell apoptosis is suggested to be via the inhibition of the phosphorylation of Rb and the expression of Bcl2, Pro-caspase3. This is based on the results of molecular docking, the gene regulatory network, and Western blot. Suppression of the CDK1 complex by linagliptin may result in the activation of the p53 signal pathway and the inhibition of the JAK–STAT signal pathway. According to a recent study, linagliptin inhibits hepatocellular cancer cells by inhibiting the protein ADORA3, and causes cell apoptosis at the G2/M phase by raising caspase3 levels.

### 2.11. Lidocaine

One of the most popular local anaesthetics in medical settings, lidocaine has been found to have a variety of uses, including the potential to treat cancer [78]. As it has been demonstrated, lidocaine exerts its multifunctional effects in analgesia, anti-inflammation, and anti-hyperalgesia through a variety of pathways, including sodium channel inhibitors and the control of G protein-coupled receptors [58]. Lidocaine also has sensitizing effects toward other chemotherapeutics, including mitomycin C and 5-fluorouracil (5-FU). In a study on the SK-MEL-2 melanoma cell line, 5-FU greatly increased the anticancer potency and apoptosis-inducing effects of lidocaine despite its low toxicity [79]. In addition to acting as a chemosensitizer, lidocaine has been shown to have inhibitory effects on a variety of cancer cells and in tumor xenograft models when used only once and at elevated concentrations. Lidocaine prevents the growth of HCC HepG2 cells by triggering apoptosis by enhancing the Bax/Bcl-2 ratio and activating caspase-3 via the ERK1/2 and p38 pathways. This suggested that lidocaine exerted its anticancer effects via cell cycle arrest at the G0/G1 phase [80]. More notably, lidocaine also sensitized cisplatin and decreased tumor development through intraperitoneal administration, indicating a combined therapy in treating HCC without expressing any toxic effects.

### 2.12. Raloxifene

Raloxifene is a benzothiophene selective estrogen receptor modulator (SERM) [59]. SERMs are a class of compounds that bind and interact with estrogen receptors and act as both agonists and antagonists for estrogen in different tissues [81]. Transcription factors, such as signal transducer and activator of transcription (STAT) proteins, are involved in signal transfer from cytokines and growth factors. Activated STAT3 enters the nucleus and induces multiple oncogene transcription, which causes cell proliferation, metastasis, and evasion from host immune system, and increases the resistance of the cell to apoptosis. Interleukin-6 (IL-6) is a cytokine which is able to induce phosphorylation of STAT3, which leads to its activation [82]. Hepatocyte repair and replication are greatly influenced by IL-6-mediated STAT3 activation, which encourages the development of hepatocarcinogenesis [83]. This process occurs by a complex pathway involving the formation of the IL-6/IL-6R$\alpha$/GP130 complex. The dimerization of this complex is the key step involved in the phosphorylation of STAT3. Raloxifene specifically inhibits IL-6 and GP130 binding. It also affects STAT3 downstream genes which induce apoptosis in cells [84].

### 2.13. Itraconazole

Itraconazole is an antifungal medicine. In recent years, further research has revealed that itraconazole has significant potential as a new anti-tumor drug and to be developed as an anti-liver cancer drug. Itraconazole blocks the abnormally active Wnt/β-catenin signaling pathway, causing cell cycle arrest and inhibiting tumor cell proliferation and metastasis. The AKT/mTOR pathway is known to be crucial in controlling biological functions in cancer cells. In HepG2 cells, itraconazole decreases the expression of p-AKT and p-mTOR and prevents the phosphorylation of the proteins PI3K and S6K, which has an impact on protein synthesis. It can be inferred from this that itraconazole suppresses HepG2 cell growth and proliferation via the PI3K/AKT/mTOR/S6K pathway [60]. The ROS pathway is activated by itraconazole, and apoptosis is induced via activating downstream caspase and PAPR proteins by balancing the ratio of pro- and anti-apoptotic proteins. By promoting the activation of the promoter caspase-8, which, in turn, activates caspase-3 and ultimately results in apoptosis, it upregulates the production of the FAS protein. Itraconazole inhibits the phosphorylation of proteins in the PI3K/AKT/mTOR/S6K signaling pathway, downregulates the Hedgehog and Wnt/-catenin signaling pathways, and downregulates the growth and proliferation of HepG2 cells, thus arresting the progression of liver cancer [85,86].

### 2.14. Clofazimine

Clofazimine is an antimycobacterial and anti-inflammatory agent which is used in treating diseases caused by mycobacterium such as leprosy, tuberculosis, and discoid lupus erythematosus [87,88]. Clofazimine increases the activity of bacterial phospholipase A2 and increases the amount of lysophospholipids which are toxic to mycobacterium [89]. Clofazimine also works by competing with menaquinone—the only quinone cofactor in mycobacteria—for the electron transported by the flavin adenine dinucleotide (FAD) of reduced NADH2. Therefore, the respiratory system oxidizes clofazimine instead of NADH. As a result, less ATP is produced during respiration [90]. Wnt signaling is a highly evolutionary conserved pathway that is important for regulating cell fate, proliferation, and migration during the development of an organism. However, in healthy adult organs, it is largely inactive. In many tissues, tumorigenesis is linked to the aberrant activation of Wnt signaling [91]. Clofazimine inhibits Wnt signaling transduction, which efficiently suppresses tumor growth. Two HCC cell lines, Hep3b and SNU398, show strong sensitivity to clofazimine, while another two lines, HepG2 and Huh7, show poor sensitivity to clofazimine [61].

### 2.15. Target Prediction

Further, the Swiss Target Prediction tool [92] was used to identify the targets of each of the repurposed drugs. The chemical SMILES of the drugs were obtained from the PubChem database and entered as input in Swiss Target Prediction, and the species was set as Homo sapiens. The tool predicted the target receptors of the drugs along with the target protein families. The results were obtained in a tabular format along with the UniProt ID, ChEMBL ID, and probability of the target receptor (Table 2).

**Table 2.** Potential repurposed drugs, their target, and receptor target class.

| Drug Name | Target | Common Name | Receptor Target Class |
|---|---|---|---|
| Pravastatin | HMG-CoA reductase | HMGCR | Oxidoreductase |
| | Neurokinin 2 receptor | TACR2 | Family A G protein-coupled receptor |
| | Norepinephrine transporter | SLC6A2 | Electrochemical transporter |
| | Dopamine transporter | SLC6A3 | Electrochemical transporter |
| | Vitamin D receptor | VDR | Nuclear receptor |
| | Thromboxane A2 receptor | TBXA2R | Family A G protein-coupled receptor |
| | Inosine-5′-monophosphate dehydrogenase 1 | IMPDH1 | Oxidoreductase |
| | Inosine-5′-monophosphate dehydrogenase 2 | IMPDH2 | Oxidoreductase |
| | Matrix metalloproteinase 1 | MMP1 | Protease |
| | Matrix metalloproteinase 8 | MMP8 | Protease |
| Atorvastatin | Cytochrome P450 3A4 | CYP3A4 | Cytochrome P450 |
| | HMG-CoA reductase | HMGCR | Oxidoreductase |
| | Histone deacetylase 6 | HDAC6 | Eraser |
| | Histone deacetylase 2 | HDAC2 | Eraser |
| | Histone deacetylase 1 | HDAC1 | Eraser |
| | Phosphodiesterase 6D | PDE6D | Phosphodiesterase |
| | Squalene synthetase | FDFT1 | Enzyme |
| | Glucocorticoid receptor | NR3C1 | Nuclear receptor |
| | Prostanoid EP4 receptor (by homology) | PTGER4 | Family A G protein-coupled receptor |
| | Phosphodiesterase 5A | PDE5A | Phosphodiesterase |

**Table 2.** *Cont.*

| Drug Name | Target | Common Name | Receptor Target Class |
|---|---|---|---|
| Simvastatin | HMG-CoA reductase | HMGCR | Oxidoreductase |
| | Norepinephrine transporter | SLC6A2 | Electrochemical transporter |
| | Neurokinin 2 receptor | TACR2 | Family A G protein-coupled receptor |
| | Dopamine transporter | SLC6A3 | Electrochemical transporter |
| | Histone deacetylase 6 | HDAC6 | Eraser |
| | Corticotropin releasing factor receptor 1 (by homology) | CRHR1 | Family B G protein-coupled receptor |
| | Histone deacetylase 1 | HDAC1 | Eraser |
| | Beta amyloid A4 protein | APP | Membrane receptor |
| | Bile acid receptor FXR | NR1H4 | Nuclear receptor |
| | 11-beta-hydroxysteroid dehydrogenase 1 | HSD11B1 | Enzyme |
| Fluvastatin | Cytochrome P450 2C9 | CYP2C9 | Cytochrome P450 |
| | HMG-CoA reductase | HMGCR | Oxidoreductase |
| | Inosine-5′-monophosphate dehydrogenase 1 | IMPDH1 | Oxidoreductase |
| | P2X purinoceptor 3 | P2RX3 | Ligand-gated ion channel |
| | Prostanoid EP4 receptor | PTGER4 | Family A G protein-coupled receptor |
| | Inosine-5′-monophosphate dehydrogenase 2 | IMPDH2 | Oxidoreductase |
| | Prostanoid EP2 receptor (by homology) | PTGER2 | Family A G protein-coupled receptor |
| | p53-binding protein Mdm-2 | MDM2 | Other nuclear protein |
| | Type-1 angiotensin II receptor (by homology) | AGTR1 | Family A G protein-coupled receptor |
| | Peroxisome proliferator-activated receptor gamma | PPARG | Nuclear receptor |
| Metformin | Thrombin | F2 | Protease |
| | Urokinase-type plasminogen activator | PLAU | Protease |
| | Histamine H4 receptor | HRH4 | Family A G protein-coupled receptor |
| | D-amino-acid oxidase | DAO | Enzyme |
| | Histamine H3 receptor | HRH3 | Family A G protein-coupled receptor |
| | Xanthine dehydrogenase | XDH | Oxidoreductase |
| | Dihydrofolate reductase (by homology) | DHFR | Oxidoreductase |
| | Integrin alpha-V/beta-3 | ITGAV ITGB3 | Membrane receptor |
| | Coagulation factor IX | F9 | Protease |
| | Neuronal acetylcholine receptor; alpha4/beta2 | CHRNA4 CHRNB2 | Ligand-gated ion channel |
| Canagliflozin | Sodium/glucose cotransporter 2 | SLC5A2 | Electrochemical transporter |
| | Sodium/glucose cotransporter 1 | SLC5A1 | Electrochemical transporter |
| | Glucose transporter (by homology) | SLC2A1 | Electrochemical transporter |
| | Phosphodiesterase 5A | PDE5A | Phosphodiesterase |
| | Adenosine A1 receptor (by homology) | ADORA1 | Family A G protein-coupled receptor |
| | Adenosine A2a receptor | ADORA2A | Family A G protein-coupled receptor |
| | Adenosine A3 receptor | ADORA3 | Family A G protein-coupled receptor |
| | Equilibrative nucleoside transporter 1 | SLC29A1 | Electrochemical transporter |
| | Adenosine kinase | ADK | Enzyme |
| | Coagulation factor VII/tissue factor | F3 | Surface antigen |

**Table 2.** *Cont.*

| Drug Name | Target | Common Name | Receptor Target Class |
|---|---|---|---|
| Pimozide | Ubiquitin carboxyl-terminal hydrolase 1 | USP1 | Enzyme |
| | Dopamine D2 receptor | DRD2 | Family A G protein-coupled receptor |
| | Potassium channel subfamily K member 2 | KCNK2 | Voltage-gated ion channel |
| | Mu opioid receptor | OPRM1 | Family A G protein-coupled receptor |
| | Delta opioid receptor | OPRD1 | Family A G protein-coupled receptor |
| | Kappa Opioid receptor | OPRK1 | Family A G protein-coupled receptor |
| | HERG | KCNH2 | Voltage-gated ion channel |
| | Serotonin 6 (5-HT6) receptor | HTR6 | Family A G protein-coupled receptor |
| | Voltage-gated T-type calcium channel alpha-1G subunit | CACNA1G | Voltage-gated ion channel |
| | Glycine receptor subunit alpha-1 | GLRA1 | Ligand-gated ion channel |
| Valproate | Peroxisome proliferator-activated receptor delta | PPARD | Nuclear receptor |
| | Free fatty acid receptor 1 | FFAR1 | Family A G protein-coupled receptor |
| | Fatty acid binding protein intestinal | FABP2 | Fatty acid binding protein family |
| | 11-beta-hydroxysteroid dehydrogenase 1 | HSD11B1 | Enzyme |
| | Fatty acid binding protein adipocyte | FABP4 | Fatty acid binding protein family |
| | Fatty acid binding protein muscle | FABP3 | Fatty acid binding protein family |
| | Aldo-keto reductase family 1 member B10 | AKR1B10 | Enzyme |
| | Peroxisome proliferator-activated receptor alpha | PPARA | Nuclear receptor |
| | Androgen Receptor | AR | Nuclear receptor |
| | Vitamin D receptor | VDR | Nuclear receptor |
| Bexarotene | Retinoid X receptor beta | RXRB | Nuclear receptor |
| | Retinoic acid receptor gamma | RARG | Nuclear receptor |
| | Retinoid X receptor gamma | RXRG | Nuclear receptor |
| | Retinoic acid receptor beta | RARB | Nuclear receptor |
| | Retinoic acid receptor alpha | RARA | Nuclear receptor |
| | Retinoid X receptor alpha | RXRA | Nuclear receptor |
| | Cytochrome P450 26B1 | CYP26B1 | Cytochrome P450 |
| | Cytochrome P450 26A1 | CYP26A1 | Cytochrome P450 |
| | Nuclear receptor ROR-gamma | RORC | Nuclear receptor |
| | Prostanoid EP4 receptor | PTGER4 | Family A G protein-coupled receptor |
| Chloroquine | Histamine H3 receptor | HRH3 | Family A G protein-coupled receptor |
| | HERG | KCNH2 | Voltage-gated ion channel |
| | Histamine N-methyltransferase (by homology) | HNMT | Enzyme |
| | Quinone reductase 2 | NQO2 | Enzyme |
| | Prion protein | PRNP | Surface antigen |
| | Muscarinic acetylcholine receptor M2 | CHRM2 | Family A G protein-coupled receptor |
| | Alpha-1d adrenergic receptor | ADRA1D | Family A G protein-coupled receptor |
| | Norepinephrine transporter | SLC6A2 | Electrochemical transporter |
| | Serotonin 2a (5-HT2a) receptor | HTR2A | Family A G protein-coupled receptor |
| | Dopamine D3 receptor | DRD3 | Family A G protein-coupled receptor |

**Table 2.** *Cont.*

| Drug Name | Target | Common Name | Receptor Target Class |
|---|---|---|---|
| Linagliptin | Muscarinic acetylcholine receptor M1 | CHRM1 | Family A G protein-coupled receptor |
| | Dipeptidyl peptidase IV | DPP4 | Protease |
| | Fibroblast activation protein alpha | FAP | Protease |
| | Cyclin-dependent kinase 4 | CDK4 | Kinase |
| | Dipeptidyl peptidase IX | DPP9 | Protease |
| | MAP kinase p38 alpha | MAPK14 | Kinase |
| | C-C chemokine receptor type 8 | CCR8 | Family A G protein-coupled receptor |
| | Tyrosine-protein kinase ABL | ABL1 | Kinase |
| | Platelet-derived growth factor receptor beta | PDGFRB | Kinase |
| | Thrombin and coagulation factor X | F10 | Protease |
| Lidocaine | Sodium channel protein type IV alpha subunit | SCN4A | Voltage-gated ion channel |
| | Serotonin 1b (5-HT1b) receptor (by homology) | HTR1B | Family A G protein-coupled receptor |
| | Dopamine D4 receptor | DRD4 | Family A G protein-coupled receptor |
| | Muscarinic acetylcholine receptor M5 | CHRM5 | Family A G protein-coupled receptor |
| | Dopamine D2 receptor | DRD2 | Family A G protein-coupled receptor |
| | Muscarinic acetylcholine receptor M4 | CHRM4 | Family A G protein-coupled receptor |
| | Cytochrome P450 2D6 | CYP2D6 | Cytochrome P450 |
| | Dopamine D1 receptor | DRD1 | Family A G protein-coupled receptor |
| | Alpha-2b adrenergic receptor | ADRA2B | Family A G protein-coupled receptor |
| | Serotonin 1e (5-HT1e) receptor | HTR1E | Family A G protein-coupled receptor |
| Raloxifene | Serotonin 2b (5-HT2b) receptor | HTR2B | Family A G protein-coupled receptor |
| | Tyrosine-protein kinase FYN | FYN | Kinase |
| | Alpha-2a adrenergic receptor | ADRA2A | Family A G protein-coupled receptor |
| | Serotonin 1b (5-HT1b) receptor (by homology) | HTR1B | Family A G protein-coupled receptor |
| | Adrenergic receptor alpha-2 | ADRA2C | Family A G protein-coupled receptor |
| | Alpha-2b adrenergic receptor | ADRA2B | Family A G protein-coupled receptor |
| | Dopamine D1 receptor | DRD1 | Family A G protein-coupled receptor |
| | Estrogen receptor alpha | ESR1 | Nuclear receptor |
| | Dopamine D2 receptor | DRD2 | Family A G protein-coupled receptor |
| | Acetylcholinesterase | ACHE | Hydrolase |
| Itraconazole | Vasopressin V2 receptor | AVPR2 | Family A G protein-coupled receptor |
| | Tyrosine-protein kinase FYN | FYN | Kinase |
| | C-C chemokine receptor type 4 | CCR4 | Family A G protein-coupled receptor |
| | Cytochrome P450 3A4 | CYP3A4 | Cytochrome P450 |
| | Cytochrome P450 51 | CYP51A1 | Cytochrome P450 |
| | Interleukin-8 receptor A | CXCR1 | Family A G protein-coupled receptor |
| | Muscarinic acetylcholine receptor M4 | CHRM4 | Family A G protein-coupled receptor |
| | Muscarinic acetylcholine receptor M5 | CHRM5 | Family A G protein-coupled receptor |
| | Sigma opioid receptor | SIGMAR1 | Membrane receptor |
| | Dopamine D3 receptor | DRD3 | Family A G protein-coupled receptor |

**Table 2.** *Cont.*

| Drug Name | Target | Common Name | Receptor Target Class |
|---|---|---|---|
| Clofazimine | Cyclophilin A (by homology) | PPIA | Isomerase |
| | Cannabinoid receptor 1 (by homology) | CNR1 | Family A G protein-coupled receptor |
| | Progesterone receptor | PGR | Nuclear receptor |
| | 6-phosphofructo-2-kinase/fructose-2,6-bisphosphatase 3 | PFKFB3 | Enzyme |
| | MAP kinase p38 alpha (by homology) | MAPK14 | Kinase |
| | Corticotropin releasing factor receptor 1 | CRHR1 | Family B G protein-coupled receptor |
| | Hepatocyte growth factor receptor | MET | Kinase |
| | Glucagon receptor | GCGR | Family B G protein-coupled receptor |
| | Translocator protein (by homology) | TSPO | Membrane receptor |
| | G protein-coupled receptor 39 | GPR39 | Family A G protein-coupled receptor |

Later, the proteins targeted by these compounds were used to create protein interaction networks using Cytoscape [93]. The interaction network is useful to understand the molecular pathways targeted by these compounds. We can see a great deal of similarity and commonality in the pathways involved in liver cancer and the pathways targeted by these compounds (Table 3).

**Table 3.** Repurposed drugs and their molecular functions affecting liver cancer.

| Drug Name | Molecular Function (Gene Ontology) |
|---|---|
| Pravastatin | Bile acid receptor activity |
| | Bradykinin receptor binding |
| | IMP dehydrogenase activity |
| | Dopamine:sodium symporter activity liver cancer |
| | JUN kinase activity |
| Atorvastatin | Bile acid receptor activity |
| | Lysophosphatidic acid receptor activity |
| | Prostaglandin receptor activity |
| | 3,5-cyclic-AMP phosphodiesterase activity |
| | Cysteine-type endopeptidase activity |
| Simvastatin | Orexin receptor activity |
| | Macrophage colony-stimulating factor receptor activity |
| | Prostaglandin-endoperoxide synthase activity |
| | Neurotrophin receptor activity |
| | PTB domain binding |
| Fluvastatin | AMP deaminase activity |
| | Bradykinin receptor binding |
| | Endothelin receptor activity |
| | IMP dehydrogenase activity |
| | Prostaglandin receptor activity |

**Table 3.** *Cont.*

| Drug Name | Molecular Function (Gene Ontology) |
|---|---|
| Metformin | Nitric-oxide synthase activity |
| | Histamine receptor activity |
| | Arginine binding |
| | G protein-coupled acetylcholine receptor activity |
| | Folic acid binding |
| Canagliflozin | JUN kinase activity |
| | Glucosylceramidase activity |
| | Ubiquitin activating enzyme activity |
| | MAP kinase activity |
| | MAP activity |
| Pimozide | Tachykinin receptor activity |
| | alpha2-adrenergic receptor activity |
| | alpha1-adrenergic receptor activity |
| | Histone kinase activity |
| | Nitric-oxide synthase activity |
| Valproate | Bile acid receptor activity |
| | Geranylgeranyl reductase activity |
| | Prostaglandin f receptor activity |
| | Prostaglandin j receptor activity |
| | Prostaglandin receptor activity |
| Bexarotene | Bradykinin receptor binding |
| | Prostaglandin d receptor activity |
| | Prostaglandin receptor activity |
| | Arachidonate 15-lipoxygenase activity |
| | DNA binding domain binding |
| Chloroquine | Alpha-adrenergic receptor activity |
| | Tachykinin receptor activity |
| | alpha1-adrenergic receptor activity |
| | G protein-coupled acetylcholine receptor activity |
| | Adrenergic receptor activity |
| Linagliptin | FBXO family protein binding |
| | Bradykinin receptor activity |
| | Insulin-activated receptor activity |
| | Platelet activating factor receptor activity |
| | Somatostatin receptor activity |
| Licodaine | Dopamine neurotransmitter receptor activity |
| | Serotonin binding |
| | Dopamine binding |
| | Adrenergic receptor activity |
| | Catecholamine binding |

**Table 3.** *Cont.*

| Drug Name | Molecular Function (Gene Ontology) |
|---|---|
| Raloxifene | Rho-dependent protein serine/threonine kinase activity |
| | Acetylcholinesterase activity |
| | Alpha-adrenergic receptor activity |
| | Serotonin binding |
| | Dopamine neurotransmitter receptor activity |
| Itraconazole | Alpha-adrenergic receptor activity |
| | G protein-coupled acetylcholine receptor activity |
| | JUN kinase activity |
| | Serotonin binding |
| | MAP kinase activity |
| Clofazimine | JUN kinase activity |
| | Cannabinoid receptor activity |
| | Orexin receptor activity |
| | Somatostatin receptor activity |
| | G protein-coupled acetylcholine receptor activity |

Using these protein interaction networks, we studied the molecular function of these networks individually for each drug using the STRING database. These drugs targeted several pathways which are common to those affected in liver cancer. Pathways such as Bile acid receptor activity, IMP dehydrogenase activity, JUN kinase activity, Nitric-oxide synthase activity, and MAP kinase activity were found to be affected by these drugs. The detailed list of molecular functions of each drug affecting liver cancer is given in Table 3.

The liver cancer pathways targeted by these drugs is a promising sign for the repurposing of these drugs to treat liver cancer.

Many of the side effects that have been recorded for all of the medicines are quite harmful for liver cancer patients and must be taken into account again before using them for treatment. The use of repurposed drugs seems to be an appealing approach, but it is necessary to consider the adverse effects associated with the drugs before prescribing them as medications for treating liver cancer. Table 4 lists all the significant side effects of these drugs.

**Table 4.** Side effects of repurposed drugs.

| Name of the Drug | Chemical Name | Mechanism of Action | Side Effect |
|---|---|---|---|
| Pravastatin | (3R,5R)-3,5-dihydroxy-7-((1R,2S,6S,8R,8aR)-6-hydroxy-2-methyl-8-{[(2S)-2-methylbutanoyl]oxy}-1,2,6,7,8,8a-hexahydronaphthalen-1-yl)-heptanoic acid | Competitive inhibition of HMG-CoA reductase to reduce cholesterol metabolism | Headache, nausea, muscle pain, rashes |
| Simvastatin | [(1S,3R,7S,8S,8aR)-8-[2-[(2R,4R)-4-hydroxy-6-oxooxan-2-yl]ethyl]-3,7-dimethyl-1,2,3,7,8,8a-hexahydronaphthalen-1-yl] 2,2-dimethylbutanoate | Competitive inhibition of HMG-CoA reductase to reduce cholesterol metabolism | Nausea, headache, memory loss, stomach pain |
| Fluvastatin | (E,3R,5S)-7-[3-(4-fluorophenyl)-1-propan-2-ylindol-2-yl]-3,5-dihydroxyhept-6-enoic acid | Competitive inhibition of HMG-CoA reductase, effect on SREBP1 pathway to reduce cholesterol metabolism | Chills, loss of appetite, muscle ache, joint pain |
| Disulfiram | N,N-diethyl[(diethylcarbamothioyl)disulfanyl]carbothioamide | Combination with Copper has cytotoxic events | Blurred vision, chest pain, confusion, nausea |

**Table 4.** *Cont.*

| Name of the Drug | Chemical Name | Mechanism of Action | Side Effect |
|---|---|---|---|
| Metformin | 3-(diaminomethylidene)-1,1-dimethylguanidine | Blocks mitochondrial respiratory chain, reducing ATP concentration | Nausea, stomach ache, loss of appetite, metallic taste in mouth |
| Canagliflozin | (2*S*,3*R*,4*R*,5*S*,6*R*)-2-[3-[[5-(4-fluorophenyl)thiophen-2-yl]methyl]-4-methylphenyl]-6-(hydroxymethyl)oxane-3,4,5-triol | Suppression of intracellular glucose uptake by HCC cells by interfering with SGLT2 and GLUT1 pathway | Indigestion, nausea, loss of appetite, trouble in breathing |
| Pimozide | 3-[1-[4,4-bis(4-fluorophenyl)butyl]piperidin-4-yl]-1*H*-benzimidazol-2-one | Inhibits stem-like cells and carcinogenesis in HCC cells | Weakness, constipation, changes in posture, dry mouth |
| Valproate | 2-propylpentanoic acid | Reduces the activity of the HDAC (histone deacetylases) gene and tumor cell differentiation | Stomach ache, tremors, headache, weight gain |
| Bexarotene | 4-[1-(3,5,5,8,8-pentamethyl-6,7-dihydronaphthalen-2-yl)ethenyl]benzoic acid | Selective inhibition of RXR that reduces angiogenesis and metastasis | Weakness, chills, weight gain, skin rash |
| Chloroquine | 4-*N*-(7-chloroquinolin-4-yl)-1-*N*,1-*N*-diethylpentane-1,4-diamine | Inhibition of receptor tyrosine kinases and mTORC1 pathway | Bleeding gums, difficulty in breathing, paralysis, nausea |
| Linagliptin | 8-[(3*R*)-3-aminopiperidin-1-yl]-7-but-2-ynyl-3-methyl-1-[(4-methylquinazolin-2-yl)methyl]purine-2,6-dione | Causes cell cycle arrest at G2/M and S phase | Trembling, sweating, confusion, difficulty concentrating |
| Lidocaine | 2-(diethylamino)-*N*-(2,6-dimethylphenyl)acetamide | Chemosensitizing effect with 5-fluorouracil increases its anticancer potency and apoptosis inducing effects | Headache, drowsiness, feeling fear, blistering at site of application |
| Raloxifene | [6-hydroxy-2-(4-hydroxyphenyl)-1-benzothiophen-3-yl]-[4-(2-piperidin-1-ylethoxy)phenyl]methanone | Inhibition of IL-6 and GP130 binding, and STAT3 genes | Hot flashes, trouble in sleeping, swollen joints, depression |
| Itraconazole | 2-butan-2-yl-4-[4-[4-[4-[[(2*R*,4*S*)-2-(2,4-dichlorophenyl)-2-(1,2,4-triazol-1-ylmethyl)-1,3-dioxolan-4-yl]methoxy]phenyl]piperazin-1-yl]phenyl]-1,2,4-triazol-3-one | Inhibition of Wnt/β-catenin signaling pathway, causing cell cycle arrest | Trouble breathing, mood changes, irregular heartbeat, increased thirst |
| Clofazimine | *N*,5-bis(4-chlorophenyl)-3-propan-2-yliminophenazin-2-amine | Blocks menaquinone, leading to reduction in ATP production | Decreased vision, bone pain, irregular heartbeat, depression |

## 3. Methods

The current review has been conducted in accordance with the PRISMA2020 guidelines. As this study does not involve any clinical or preclinical data, a Systematic Review Registration is not required for this review [94]. Based on the papers published between 1998 and 2022, a thorough analysis of the data revealing the significance of drug repurposing in liver cancer was conducted. The data for our investigation came from PubMed, ScienceDirect, and JSTOR using the following associated keywords in combination: drug repurposing, liver cancer, medication repurposing, diabetes, cancer, statins, anti-alcoholism, chronic psychosis, epilepsy, and bipolar disorder.

The eligibility of the studies was defined based on the following inclusion criteria: (i) studies published in the English language were chosen; (ii) original studies elucidating the effects of previously existing drugs and their interaction with receptors that may be a potential target for treating liver cancer; and (iii) research papers and clinical trial studies were chosen for their authenticity. The exclusion criteria were (i) studies published in languages other than English; (ii) unavailability of the full text of the study; (iii) studies found to be irrelevant once the full text is obtained; (iv) lack of clarity; and (v) lack of rigor.

## 4. Conclusions

Liver cancer is one of the most common malignancies with a high mortality rate. Identifying treatment options with minimal toxicity is essential for an effective therapeutic outcome; one such approach is drug repurposing. Drug repurposing is a practical approach to finding approved drugs for alternate diseases. The main advantages of this approach include quick processing time, reduced cost for drug development, and a less tedious approval process. The study identifies 14 drugs from different pathologies, targeting different classes of drug receptors in various diseases. The possibility of repurposing these drugs to treat liver cancer has been discussed. Computational techniques such as molecular docking and molecular dynamic simulation can be paired with this approach to study the most potent drug.

These repurposed drugs have great potential for treating liver cancer, but their adverse effects must also be considered. The side effects for each of the repurposed drugs in this study have been mentioned in Table 4. Additionally, despite these drugs being approved by regulatory authorities, they must undergo clinical trials to study their effect on different pathologies. Drug repurposing is an attractive alternative to the slow-paced traditional drug discovery process. It provides an opportunity to utilize previously approved drugs for targeting receptors for various diseases outside the scope of the original medication.

**Author Contributions:** Conceptualization, R.J.M., D.P.K. and A.M.P.; Data curation, A.T.; Methodology, R.J.M., M.A., K.A., D.P.K. and A.M.P.; Resources, P.N.; Supervision, R.J.M.; Writing—original draft, R.J.M., M.A., K.A., A.T., Q.A.R.H., T.V.S. and P.N.; Writing—review & editing, R.J.M., M.A., K.A., A.T., Q.A.R.H., T.V.S., P.N., D.P.K. and A.M.P. All authors have read and agreed to the published version of the manuscript.

**Funding:** No funding was received for the current work.

**Institutional Review Board Statement:** Not applicable.

**Informed Consent Statement:** Not applicable.

**Acknowledgments:** We acknowledge the NPDF fellowship and support given to Angamba Meetei Potshangbam by DST SERB.

**Conflicts of Interest:** The authors declare no conflict of interest.

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
