# Peer review of "A Systematic Review of Molecular Pathway Analysis of Drugs for Potential Use in Liver Cancer Treatment"

_ddc, doi:10.3390/ddc2020013_

Round 1
Reviewer 1 Report
In the manuscript (ID: ddc-2214977), the authors used databases like PubMed, Science Direct and JSTOR for literature mining and the PRISMA 2020 systemic review guideline to identify drugs that have been repurposed for liver cancer. The authors describe sorafenib as first line therapy for liver cancer. What about other drugs used for treatment of liver cancer? It should be much more discussed in the section 1.2. On the Figure 4 all the chemical structures of compounds should be the same size. In the text of the manuscript there is a lack of systematicity because some of compounds have their IUPAC name and some not. At the end in Table 3 all compound have their IUPAC names, so it can be removed from text. The text should be carefully checked. After section 4.14 the next step of the approach is target prediction what should be separated as next section. Not all references are included e.g. the lack of Swiss Target Prediction Tool reference. The two first references 1 and 2 are the same so th references should be carefully checked. The Figure 5 is unreadable so it should be removed. Why in Table 1 sorafenib is included among other repurposed drugs? Is it possible to find a link between targeted protein and molecular function of the compound?
Author Response
Response to Reviewer
Manuscript ID: ddc-2214977
Submission Title: in silico analysis of repurposed drugs in Liver Cancer
Dear Editor,
We thank the respected reviewers for their insightful comments and constructive feedback on our manuscript. In order to improve our work, we have carefully examined and incorporated all of the suggested changes. We value the reviewers' efforts because their suggestions significantly raised the paper's overall quality. The changes made in the revised manuscript are marked with yellow colour.
The answers to the reviewer comments are as follows:
Reviewer 1
In the manuscript (ID: dde-2214977), the authors used databases like PubMed, Science Direct and JSTOR for literature mining and the PRISMA 2020 systemic review guideline to identify drugs that have been repurposed for liver cancer. The authors describe sorafenib as first line therapy for liver cancer. What about other drugs used for treatment of liver cancer? It should be much more discussed in the section 1.2. On the Figure 4 all the chemical structures of compounds should be the same size. In the text of the manuscript there is a lack of systematicity because some of compounds have their IUPAC name and some not. At the end in Table 3 all compounds have their IUPAC names, so it can be removed from text. The text should be carefully checked. After section 4.14 the next step of the approach is target prediction what should be separated as next section. Not all references are included e.g., the lack of Swiss Target Prediction Tool reference. The two first references 1 and 2 are the same so the references should be carefully checked. The Figure 5 is unreadable so it should be removed. Why in Table 1 sorafenib is included among other repurposed drugs? Is it possible to find a link between targeted protein and molecular function of the compound? The tile is not accurate and precise, it needs reviewing.
- The authors describe sorafenib as first line therapy for liver cancer. What about other drugs used for treatment of liver cancer? It should be much more discussed in the section 1.2.
Answer: Authors welcome this suggestion and changes have been made accordingly at page numbers 2 & 3 in section 1.2.1.
- On the Figure 4 all the chemical structures of compounds should be the same size.
Answer: The Authors have made changes according to the suggestion.
- At the end in Table 3 all compounds have their IUPAC names, so it can be removed from text.
Answer: Authors welcome this suggestion and have removed the IUPAC from the text.
- After section 4.14 the next step of the approach is target prediction what should be separated as next section.
Answer: Authors welcome this suggestion and have created a new section – 4.15 to separate Target Prediction.
- Not all references are included e.g., the lack of Swiss Target Prediction Tool reference. The two first references 1 and 2 are the same so the references should be carefully checked
Answer: References have been added for Swiss Target Prediction Tool and Cytoscape [83], [84]. Reference 1 has been removed and duplicity of all references has been checked.
- Figure 5 is unreadable so it should be removed
Answer: Figure 5 is a collage of many interactions networks. Since the networks are huge it is not possible to view nodes. This figure is depicting the results obtained after following the gene-interaction methodology. This figure is important for the manuscript.
- Why in Table 1 sorafenib is included among other repurposed drugs? Is it possible to find a link between targeted protein and molecular function of the compound?
Answer: Authors thank the reviewer for pointing out the error. Sorafenib has been removed from Table 1. Yes, it is possible to find a link between targeted protein and molecular function. In the current review we have extracted the common molecular functions of the entire focused network so as to understand the overall effect of the drug on the body.
- The tile is not accurate and precise, it needs reviewing.
Answer: As suggested, the title has been revised.
Reviewer 2 Report
ddc-2214977, Insilico analysis of repurposed drugs in Liver Cancer
The manuscript presents an interesting research. It has an average value, but it could be published after some important improvements.
The first section of the introduction (rows 23 up to 135) focus on hepatic cancer, but this section is too long and out of the real objective of the paper. It would be much better to discuss what drugs are already in use (approved) for this type of cancer. What are problems with those and why do we need some other new drugs. The author tried something like this on row 84, but they should expand to more drugs.
I would remove the figure 1. It presents a local (UK) trend and shows what is already a common knowledge.
The title need corrections. It should be “in silico”. Also, the drugs are not repurposed. They have the potential to be used in the future. In my opinion a repurposed drug is a drug that is approved for a new indication, other than the ones intended in the drug development stage. In this case the drug listed in the manuscript would have to be approved for liver cancer treatment.
The manuscript looks more like a review, and less an original work. The authors should clearly present the methods used and their original data. In my view, there are little original data to call this paper original. I advise the authors to correct its structure and present it as a review, or as they call it “systematic review”
There are many editing errors. See as example: Aspergillus terreus should be italics. Check all the paper. See also row 230, “existing…”
Author Response
Response to Reviewer
Manuscript ID: ddc-2214977
Submission Title: in silico analysis of repurposed drugs in Liver Cancer
Dear Editor,
We thank the respected reviewers for their insightful comments and constructive feedback on our manuscript. In order to improve our work, we have carefully examined and incorporated all of the suggested changes. We value the reviewers' efforts because their suggestions significantly raised the paper's overall quality. The changes made in the revised manuscript are marked with yellow colour.
The answers to the reviewer comments are as follows:
Reviewer 2
The manuscript presents an interesting research. It has an average value, but it could be published after some important improvements.
The first section of the introduction (rows 23 up to 135) focus on hepatic cancer, but this section is too long and out of the real objective of the paper. It would be much better to discuss what drugs are already in use (approved) for this type of cancer. What are problems with those and why do we need some other new drugs. The author tried something like this on row 84, but they should expand to more drugs.
I would remove the figure 1. It presents a local (UK) trend and shows what is already a common knowledge.
The title needs corrections. It should be "in silico". Also, the drugs are not repurposed. They have the potential to be used in the future. In my opinion a repurposed drug is a drug that is approved for a new indication, other than the ones intended in the drug development stage. In this case the drug listed in the manuscript would have to be approved for liver cancer treatment.
The manuscript looks more like a review, and less an original work. The authors should clearly present the methods used and their original data. In my view, there are little original data to call this paper original. I advise the authors to correct its structure and present it as a review, or as they call it "systematic review"
There are many editing errors. See as example: Aspergillus terreus should be italics. Check all the paper. See also row 230, "existing..."
- The first section of the introduction (rows 23 up to 135) focus on hepatic cancer, but this section is too long and out of the real objective of the paper. It would be much better to discuss what drugs are already in use (approved) for this type of cancer. What are problems with those and why do we need some other new drugs. The author tried something like this on row 84, but they should expand to more drugs.
Answer: Authors are thankful for this suggestion and have made changes in the introduction section from Pages 1-5.
- I would remove the figure 1. It presents a local (UK) trend and shows what is already a common knowledge.
Answer: Authors welcome this suggestion and have removed figure 1.
- The title needs corrections. It should be "in silico". Also, the drugs are not repurposed. They have the potential to be used in the future. In my opinion a repurposed drug is a drug that is approved for a new indication, other than the ones intended in the drug development stage. In this case the drug listed in the manuscript would have to be approved for liver cancer treatment.
Answer: Authors welcome the point made by the reviewer and have amended the relevant changes in the title.
- The manuscript looks more like a review, and less an original work. The authors should clearly present the methods used and their original data. In my view, there are little original data to call this paper original. I advise the authors to correct its structure and present it as a review, or as they call it "systematic review".
Answer: Authors welcome this suggestion. The study conducted was a systematic review, and the PRISMA guidelines were strictly followed. The authors have revised the manuscript based on the suggestion provided
- There are many editing errors. See as example: Aspergillus terreus should be italics. Check all the paper. See also row 230, "existing..."
Answer: Thank you for this suggestion. The paper is revised with relevant corrections.
Row 230 has also been checked and corrected. Thank you.
Round 2
Reviewer 1 Report
In the manuscript ddc-2214977 the authors correct almost everyting according suggestions. However, the Figure 4 should be removed from the manuscript because it's unreadeable. The authors include major informations of pathways in Table 3.
Author Response
We thank the reviewer for the suggestion. Figure 4 has been removed in the revised manuscript.
Reviewer 2 Report
The authors made the suggested changes and improved the quality of their paper.
Author Response
The authors thank the reviewer for taking the time to evaluate our manuscript and for providing valuable feedback. Your comments and suggestions have been instrumental in helping us to improve the quality of our manuscript.